# Thermal Behavior in Glass Houses through the Analysis of Scale Models

**Patricia Aguilera-Benito [1,*] , Sheila Varela-Lujan [2] and Carolina Piña-Ramirez [3]**

1   Departamento de Tecnología de la Edificación, Escuela Técnica Superior de Edificación, Universidad Politécnica de Madrid, 28040 Madrid, Spain

2   Escuela Técnica Superior de Edificación, Universidad Politécnica de Madrid, 28040 Madrid, Spain; sheila.varela.lujan@alumnos.upm.es

3   Departamento de Construcciones Arquitectónicas y su Control, Escuela Técnica Superior de Edificación, Universidad Politécnica de Madrid, 28040 Madrid, Spain; carolina.pina@upm.es

*   Correspondence: patricia.aguilera@upm.es

**Abstract:** Reducing energy expenditure in the construction sector requires the implementation of passive strategies in buildings. In Spain, consumption is centered on air conditioning systems associated with the demand for the building's thermal envelope. A critical point of the enclosures is represented by glazed holes where much of the energy that is consumed is lost; however, homes increasingly tend to have large window openings due to the comfort and visual well-being they provide to users. In this study, we focus on an extreme case, analyzing a fully glazed house in its four orientations. It is necessary to evaluate the most energy efficient passive strategy for this type of construction. The results are based on the temperature analysis obtained during the monitoring of two scale models of a glass house. The results indicate that solar control foil glasses perform better in warmer weather stations. Regarding the cantilever installation, it influences the interior temperature and the central hours of the day, mitigating the increase in temperature as well as slowing the nighttime cooling.

**Keywords:** glass facade; passive strategies; solar control sheet; cantilevers; thermal analysis

## 1. Introduction

Environmental problems tend to be complicated by population growth and economic growth in markets that leads to an increase in the global demand for energy; people also have greater purchasing power and are increasingly demanding with respect to their well-being and comfort criteria [1]. Energy has been an essential element for the subsistence and development of civilization; however, the predominant energy model was mainly based on the use of fossil fuels, causing serious problems for humanity and the environment [2]. Currently, renewable energies are partially replacing fossil fuels, albeit remaining a developing scenario today. According to the International Energy Agency, only 13% of the total supply of primary energy in the world is produced from renewable energy sources [3]. This is an incredibly low percentage that must be increased in the coming years to reach the Sustainable Development Goals (SDG) established by the United Nations General Assembly for the next 15 years.

Spain is one of the countries in the European Union with the greatest potential for the use of renewable energy in buildings, particularly solar energy. This is due to the number of hours of sunshine (an average of approximately 2500 h of sunshine per year) it receives and the warm temperatures typical of a Mediterranean climate (temperatures remain on average every month above 20 °C but present seasonal variation) [4–6]. By incorporating bioclimatic design criteria, it is possible to generate buildings with almost zero consumption, moving us closer to what has been called sustainable architecture (i.e., the design of buildings that generate zero impact on the environment) [7]. It has been

confirmed that these criteria are not sufficient and progress must be made toward the construction of buildings that generate positive impacts. Buildings designed with the criteria of regenerative sustainability consider, in addition to energy efficiency and the use of renewable and/or alternative energies, the improvement of the health and well-being of users [8,9]. Bioclimatic architecture is a benchmark on which to consolidate this new concept as it considers the use of not only the sun as an alternative energy but also sunlight as a source of health and other benefits for people.

Glass facades benefit from both heat and light energies provided by the sun, representing a renewable energy source that does not need any equipment or system to be transformed from a primary energy to a final energy [10–12]. The composition of the thermal envelope is very important when quantifying the demand that building requires. For this reason, there are numerous studies that analyzed the thermal behavior of different facade configurations, although few were based on the thermal analysis of glass envelopes in residential use [13–18]. This type of facade is more associated with tertiary uses where an aesthetic or corporate image is sought rather than a positive energy rating [19].

There are numerous studies that have analyzed different aspects of the facades of office buildings such as the climatic zone and the orientation and dimensions of the openings but few investigations have been focused on buildings for residential use, representing a place where we spend long periods of time [20]. The World Health Organization indicated that we spend approximately 90% of time inside buildings and it is estimated that two-thirds of that time is spent inside the home [21]. In addition, considering the COVID-19 pandemic that we are currently experiencing, this time has been increased by having to carry out a large number of activities inside the home [22].

Glazed facades have been projected from technical studies; however, if this configuration is not well designed, it can be detrimental due to major energy consumption. The type and characteristics of the glass administered for use in each building must be analyzed in detail for each of the orientations and in consideration of the climatic zone in which the building is to be located [23]. For this reason, new passive strategies are being studied that reduce the increase in energy demand in these types of glass houses by increasing the glazed surfaces in the building envelope, thereby favoring the health and well-being of users by benefiting from natural light.

## 2. Methods

This section describes the passive strategies analyzed in other studies from a theoretical perspective. It also describes the monitoring that was conducted on the scale models where the experimental analyses were determined and describes the technical specifications of the glass of the different facades and the overhangs. Their measured parameters are described. This monitoring was developed by referring to previous studies [24–28].

### 2.1. Preliminary Theoretical Analysis of Passive Strategies in Buildings

The composition of the thermal envelope is very important when quantifying the energy demand of the building. For this reason, there are numerous studies that analyze the thermal behavior of different facade configurations.

For example, the authors Friess and Raksan analyze the effects of radiation, convection and heat conduction through walls, and roofs and windows, as well as assess the most efficient natural ventilation techniques in different buildings, obtaining in all cases optimized energy configurations with reductions of up to 30% in residential buildings and 79% in office buildings [29].

Chen, Yang, and Wang analyzed the parameters that affect solar energy gain in the for high-rise buildings. The most decisive parameters being the solar heat gain coefficient, the gap/shell ratio, the external obstruction angle, and the projection fraction [30].

In the study conducted by Kumar and Babu, an office building in India is analyzed considering the spectral properties of different glass materials. The results indicate that in terms of the daylight factor, green reflective glass placed in the south orientation performs

best in the summer and bronze colored glass placed in the north orientation performs best during the winter [31]. Additionally, it should be noted that the importance of a detailed analysis by orientation for the choice of the most suitable glass in each case is fundamental, as concluded in the research by Cuce and Banihashemi [32].

Conversely, the work of Aste, Buzzetti, Del Pero, and Leonforte indicates that energy balances in an office building in different climate zones demonstrate that there is a negative impact of solar gains. Thus, the most reasonable solution to reduce annual energy consumption is to use external solar shading devices or utilize solar control glazing [33].

In any case, glass is a material that has developed exponentially in recent years and has evolved with new technologies. For example, Giménez's study thermally analyzes a glass with a circulating water chamber that reduces cooling consumption by attenuating solar gains in the glazing with minimum consumption in the recirculation pump [34]. Similarly, the Pérez-Pujazón study utilizes this same typology of glazing including an air conditioning and/or heating system of renewable energy to justify a viable alternative for near-zero energy buildings [35]. Research has also been conducted concerning glazing with phase change materials in the chambers between the panes, leading to better results than traditional glass as it partly prevents the transmission of solar gains to the interior [36].

Other studies consider the economics of new glass materials such as Fazel's studies on the installation and maintenance costs of glass envelopes. Fazel indicates that new smart, high-technology windows have a great impact on reducing heating and cooling costs in buildings; however, the initial costs of these windows are excessively high. For this reason, he proposes two new methods to build low-cost smart windows. The first method is based on the pressure variations of a gas that depend on temperature fluctuation and the second is a two-phase liquid fluid with different coefficients of thermal expansion [37]. In addition, a study conducted in residential buildings in Catalonia demonstrates that the most expensive measures do not always ensure the best results in terms of energy savings. Therefore, the best materials or most innovative and therefore most expensive systems do not always lead to the most optimal solutions [38]. Finally, the study "Economic assessments of passive thermal rehabilitations of dwellings in Mediterranean climate" presents an energy and economic analysis to evaluate the profitability of energy renovation operations on the envelope of a block of flats. This work concludes that the most cost-effective scenario is the one that combines a facade with insulation above the regulatory limit, double-glazed windows with low emissivity, and aluminum frames with thermal break and solar protection with movable shading elements [39].

Based upon the literature consulted, it is concluded that in buildings the glazed opening and its energy transmission properties are an element of great potential for both energy saving and energy cost control. It is also confirmed that local conditions, orientation, percentage of blind and glazed facade, glazing characteristics, and passive protections are determining properties in both construction and energy aspects. These aspects are considered fundamental, absent of the necessity to install highly innovative systems or high installation and maintenance costs.

### 2.2. Experimental Model

The reference house for this study is the Farnsworth House that is rectangular in shape, not in contact with the ground, and has a west-facing entrance porch [40,41].

For the development of the experimental phase, we established two 1:6 scale models to be able to compare in equal climatic conditions the energy behavior of different glass enclosures. The choice of the scale was determined after calculations were conducted following the book *Building Ventilation Theory and Measurement* [42] that discusses the theory of similarities used in scale models with the aim of making their behavior as close as possible to demonstrate how they would behave in a real situation. The three similarities of geometry, kinetics, and dynamics should be achieved for the scaled model to the full-scale building. Geometric similarity is the easiest of these criteria to meet; simply scale down the linear dimensions of the full-scale building equally for three dimensions. In this paper, the

scaled model is one-sixth of the size of the prototype building. For kinematic and dynamics similarity, the ratios of the fluid velocities and accelerations must be equal. In reference to similarity theory, a scale model for buoyancy-driven natural ventilation will replicate the kinematic boundary conditions of the prototype if the Prandtl, Reynolds, and Archimedes numbers are similar for both cases. To meet these properties, the scale models have natural ventilation openings [43].

The scale models represent an open-plan house glazed all over its facade perimeter with a 1.15 m long porch in its west orientation. Inside the house, there is only a central volume slightly displaced toward the north face, simulating the spaces where the toilets and/or bathrooms would be located (Figures 1 and 2).

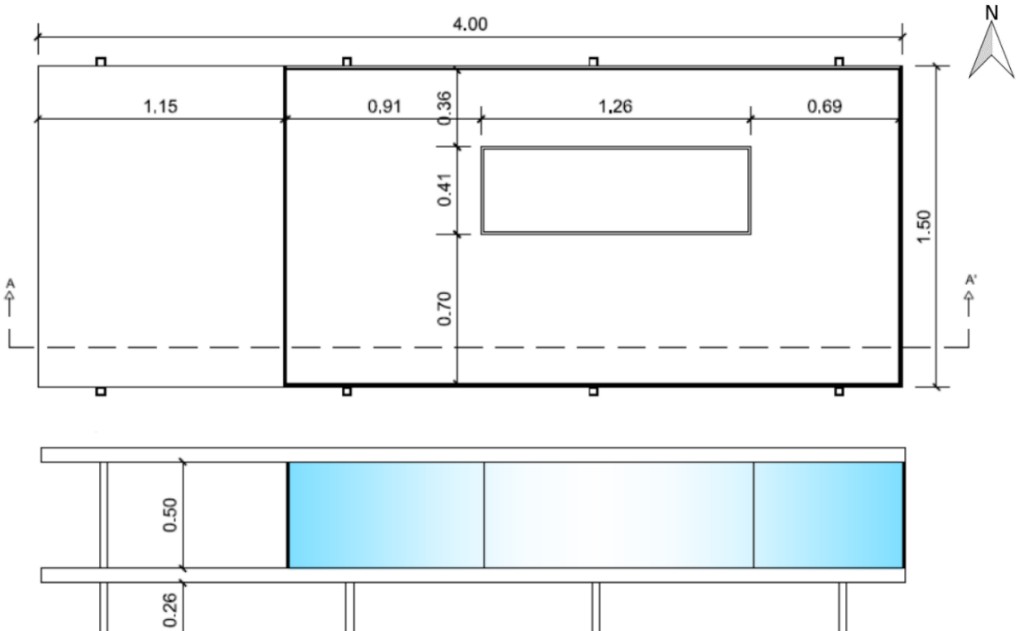

**Figure 1.** Plan and section of the scale models.

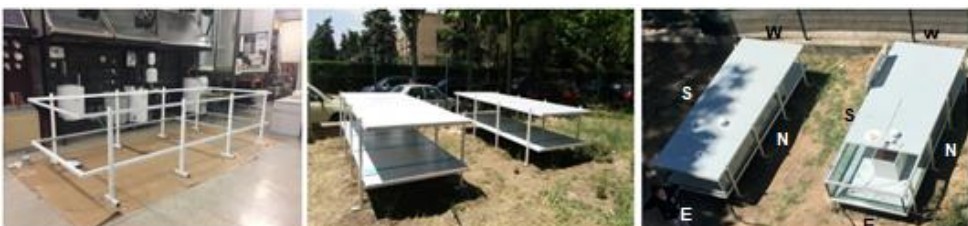

**Figure 2.** Construction of the scale models.

Both scale models were placed inside the complex of the Escuela Técnica Superior de Edificación de Madrid of the Universidad Politécnica de Madrid. The scale models were separated by 2.00 m from each other to avoid interference between their own shadows.

In addition, the shade provided by nearby deciduous trees was studied. The following results were obtained from the analyses where the scale models were shaded from dawn until the specified times (Table 1).

**Table 1.** Hours in the shadow scale models.

| Month | Reference Model (from Dawn to:) | Improved Model (from Dawn to:) |
|---|---|---|
| March and September | 6:30 a.m. | 9:00 a.m. |
| April and August | 7:30 a.m. | 9:45 a.m. |
| May and July | 8:30 a.m. | 10:15 a.m. |
| June | 9:30 a.m. | 10:30 a.m. |

As presented, the time difference between the beginning of the shadow and the end between one model and another is around 2 h. Furthermore, from 10:30 a.m. and onwards, no model receives shadow, thereby it does not affect the midday time slot that is most important for this analysis. For this reason, this data is taken as a reference to make the appropriate comparisons in the analyses and thus be able to accurately analyze the results obtained.

The scale models were built with a steel perimeter structure as the upper and lower frame, supported by eight metal profiles comprising the pillars of the building. The lower and upper floors were solved with sandwich panel pieces. The glass panels of both models were placed on the profiles without direct contact on neoprene bands to isolate the thermal bridge between the glass and the metal of the profiles, as shown in Figure 3.

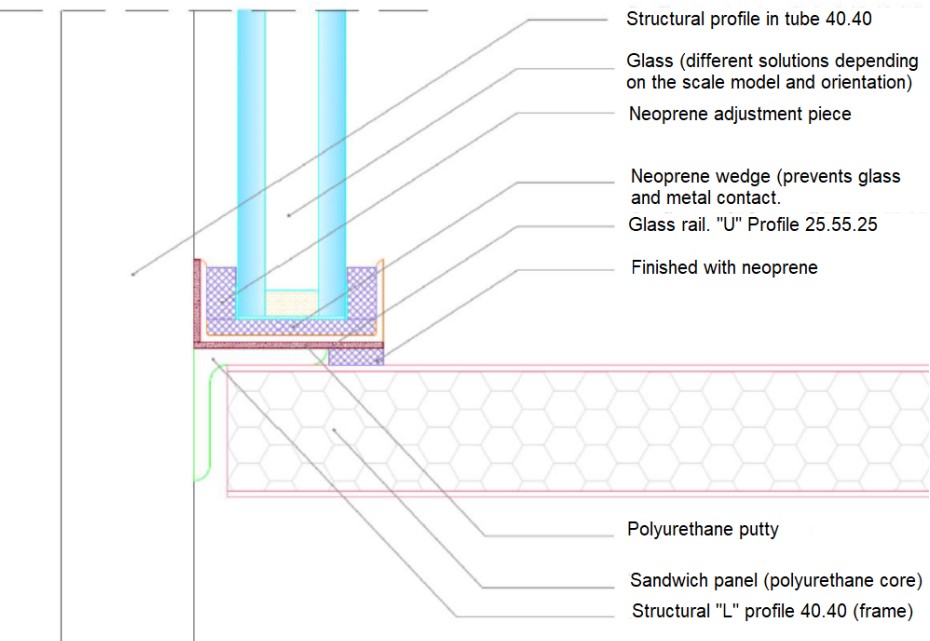

**Figure 3.** Construction details: glass support profiles.

Two scale models were executed with the same geometric and constructive criteria, and two passive strategies were implemented and experimentally analyzed.

In the first case, in the reference model, simple glass panels were installed, comprising of 6 mm thick monolithic glazing in all four orientations. In the improved model, 6 + 16 + 6 mm double glazing was installed in south, east, and west orientations with a solar control sheet on face 2 of the glazing, whereas in the north orientation a conventional 6 + 16 + 6 mm double glass was installed [44]. The technical characteristics of the installed glass panels are reflected in Table 2.

**Table 2.** Technical specifications of the glass panels installed in the models.

| Model | Orientation | Glass Type | Total Solar Transmission SHGC "g" (%) | Direct Solar Transmission (%) | U Value (ISO 10292) (W/m²·K) |
|---|---|---|---|---|---|
| Reference | All | 6 mm single glass (clear float glass) | 86.0 | 89.0 | 5.7 |
| Improved | North | Double glass (conventional DA) | 77.1 | 81.9 | 2.7 |
| | South, east, and west | Double glass + SCL (SunGuard Solar Neutral 67) | 58.8 | 61.0 | 2.6 |

As a second passive strategy, overhangs were designed with the intention of protecting the interior of the house from sunlight in the 4 months with the highest solar elevation angle. From the calculations conducted, the one minimizing the maximum solar radiation inside the building in the maximum possible period was selected, obtaining a 1.80 m long overhang that was installed for a few summer months in the south orientation for both of the scale models (Figure 4).

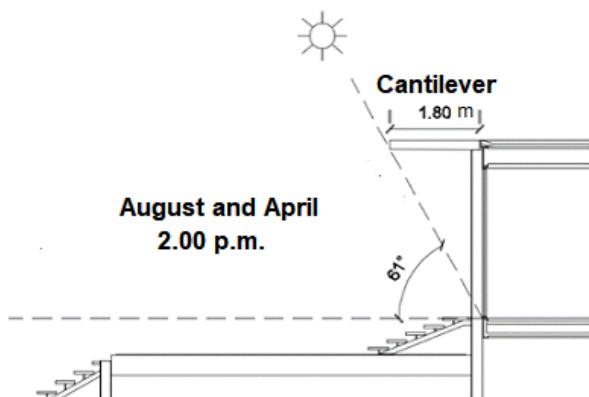

**Figure 4.** Cantilever designed according to the angle of the sun.

Thirty simulations were carried out in a previous study [45] and the cantilever designed for 21 April and 21 August at 14:00. with an angle of incidence of solar radiation with the horizontal of 61° was selected.

*2.3. Monitoring Model*

To validate the research and obtain data for all seasons, we monitored the scale models for a full year that ran from July 2017 to August 2018 [46–48]. The sensors were positioned according to a previous analysis and in compliance with ISO 7726:2002 [49]. Positioning tests were performed and data was collected for analysis in order to gather useful data.

The positioning of the sensors was established in both models equally in terms of location but not in terms of the number of thermocouples placed on the facades to measure the surface temperature. In the reference scale model, four thermocouples were placed on the exterior surface of the glass in each orientation because the temperature difference between the inside and outside was negligible. In the improved model, 8 thermocouples were placed inside and outside. In addition, a thermo-hygrometer was positioned inside each of the scale models to measure the interior ambient temperature (Table 3 and Figure 5).

**Table 3.** Measuring instruments in each of the scale models.

| Model | Units | Type of Sensor | Position |
|---|---|---|---|
| Reference | 4 | Thermocouple type K | One in each orientation |
| | 1 | OPUS 20 + $CO_2$ thermo-hygrometer | One located inside the scale model |
| Improved | 8 | Thermocouple type K | Two in each orientation located inside and outside the glass |
| | 1 | OPUS 20 + $CO_2$ thermo-hygrometer | One located inside the scale model |

The established nomenclature is, the first letter referring to the model in which the sensor is located (reference model R or modified model M). The next three letters refer to the unit of measurement of the sensor (tem = temperature) and the number indicates the positioning of the sensor within each of the scale models.

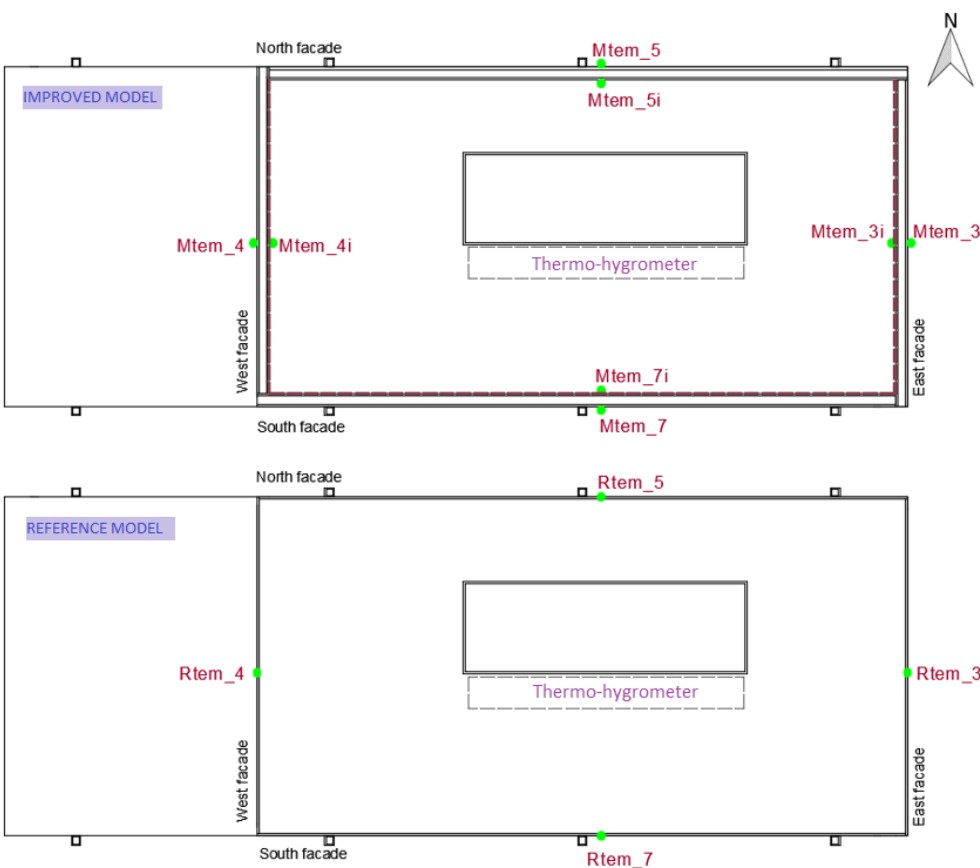

**Figure 5.** Positioning of the sensors used to measure surface and ambient temperature.

In addition to the previously detailed sensors, data were available for the external environmental climatological variables during the monitoring period from 1 August 2017 to 31 July 2018. These climatological data were obtained from the Spanish State Meteorological Agency (AEMET) that periodically provided us the data from the meteorological station closest to the scale models (i.e., the Ciudad Universitaria de Madrid).

The readings of the type K thermocouples were collected in a datalogger every 2 min and later analyzed as 10 min average values, correcting the values with the errors collected in a previous calibration of the sensors as established in the UNE-EN ISO 7726 standard on "Ergonomics in thermal environments: instruments for measuring physical quantities" [49].

## 3. Results and Discussion

In this section, we describe the results obtained from the thermocouples installed in the two scale models that collected data throughout the entire monitoring period of a

full year. The data were analyzed in the most representative period of each season. The temperatures analyzed were the interior ambient temperatures of the scale models and the surface temperatures of the glass panels in each of the scale models.

### 3.1. Experimental Data of Ambient Temperatures

The most stable period was chosen within each season. The summer parameters were recorded in two different years (2017 and 2018) and it was important that the external conditions during both selected periods were as similar as possible to ensure the samples were comparable. For this reason, the outdoor ambient temperatures were compared during the summer season and we calculated the deviation in temperatures between 2017 during which the scale models had no overhangs and 2018 during which the scale models had overhangs. The temperature records were then averaged and the three consecutive days were found in which the average temperature in the period without a cantilever (2017) deviated only 2% from the period with a cantilever (2018). Thus, it could be considered that the monitoring with and without a cantilever was conducted simultaneously. In the remaining season, three consecutive days were selected during which the average temperature difference was the smallest possible value from the average value calculated at each station. Therefore, in the analysis of the behavior of the models, the three days chosen for each period were named days 1, 2, and 3.

The first analysis was conducted on the interior ambient temperatures in the summer season. As shown in Figure 4, the incorporation of cantilevers in the scale models led to a decrease in the interior temperature of up to 6 °C during the hours of greatest solar radiation (approximately 12:00pm–4:00pm). In addition, the minimum temperatures reached inside were higher in the models with overhangs. This is due to the fact that when the temperature decreased outside, the overhang slowed heat dissipation.

The thermal jump between the maximum and minimum temperatures inside the scale models was smaller when overhangs were present. This increase of up to 8 °C would lead to a higher air conditioning consumption due to the greater temperature range throughout the day (Figures 6 and 7).

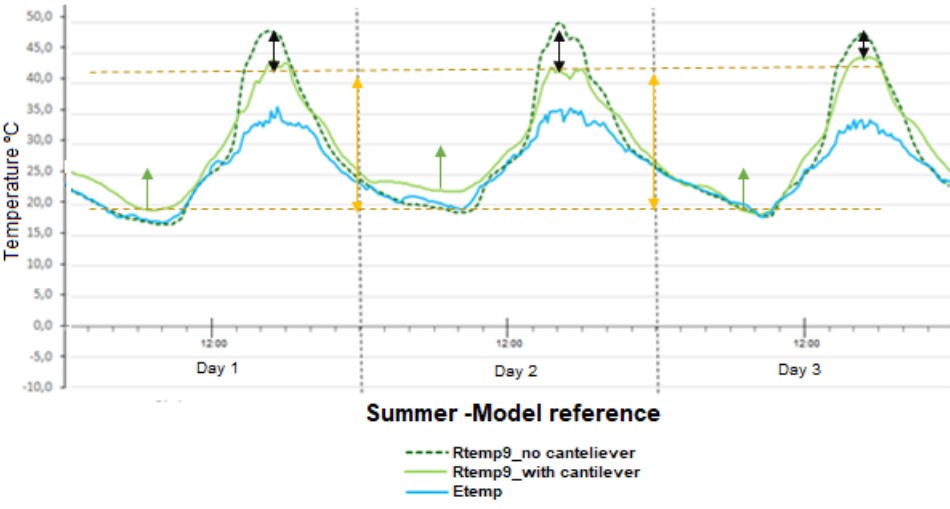

**Figure 6.** The model reference of indoor and outdoor ambient temperatures during the summer season.

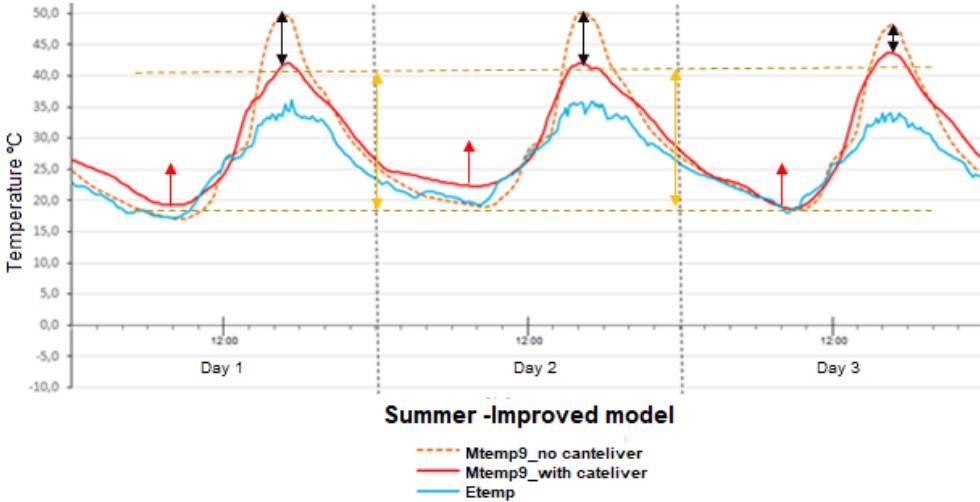

**Figure 7.** Improved model of indoor and outdoor ambient temperatures during the summer season.

In order to quantify the effect of the cantilever on inside temperatures, we performed a statistical study on the data using a simple regression model to compare the inside and outside temperatures before and after the installation of the cantilevers. The coefficient of correlation between the inside and outside temperatures is close to 1 (0.96) in the case of the models without a cantilever.

In the models with cantilevers, the correlation coefficient is lower (0.71 and 0.64), demonstrating a lower correlation between indoor and outdoor temperatures. Consequently, the overhang in the two models (with simple glass and improved glass) greatly influenced (in relation to the installation of glass) the interior temperature in the central hours of the day, mitigating the increase in temperature and slowing nighttime cooling (Table 4).

**Table 4.** Linear regressions between indoor and outdoor temperatures during the summer season.

|  | Model | r (Tª Inside/Tª Outside) | $R^2$ (%) (Tª Inside/Tª Outside) |
|---|---|---|---|
| No Cantilever | Reference | 0.961914 | 92.527931 |
|  | Improved | 0.961174 | 92.385462 |
| With Cantilever | Reference | 0.718107 | 51.567833 |
|  | Improved | 0.643541 | 41.414518 |

It should be noted that in autumn and winter, the thermal wave during daytime periods was delayed in the improved glass model in relation to the simple glass model, with the opposite occurring in spring (Figure 7, identified with a black box). In addition, the difference between the maximum values of both scale models was 6.6 °C in autumn and 5.7 °C in the winter with no significant difference observed in the spring. These results were attributed to the solar control sheet that in autumn and winter did not allow complete entry of the incident solar radiation, whereas in the spring the increase in the number of hours of sunshine caused the glass to overheat more quickly, thus reversing the process (Figure 8).

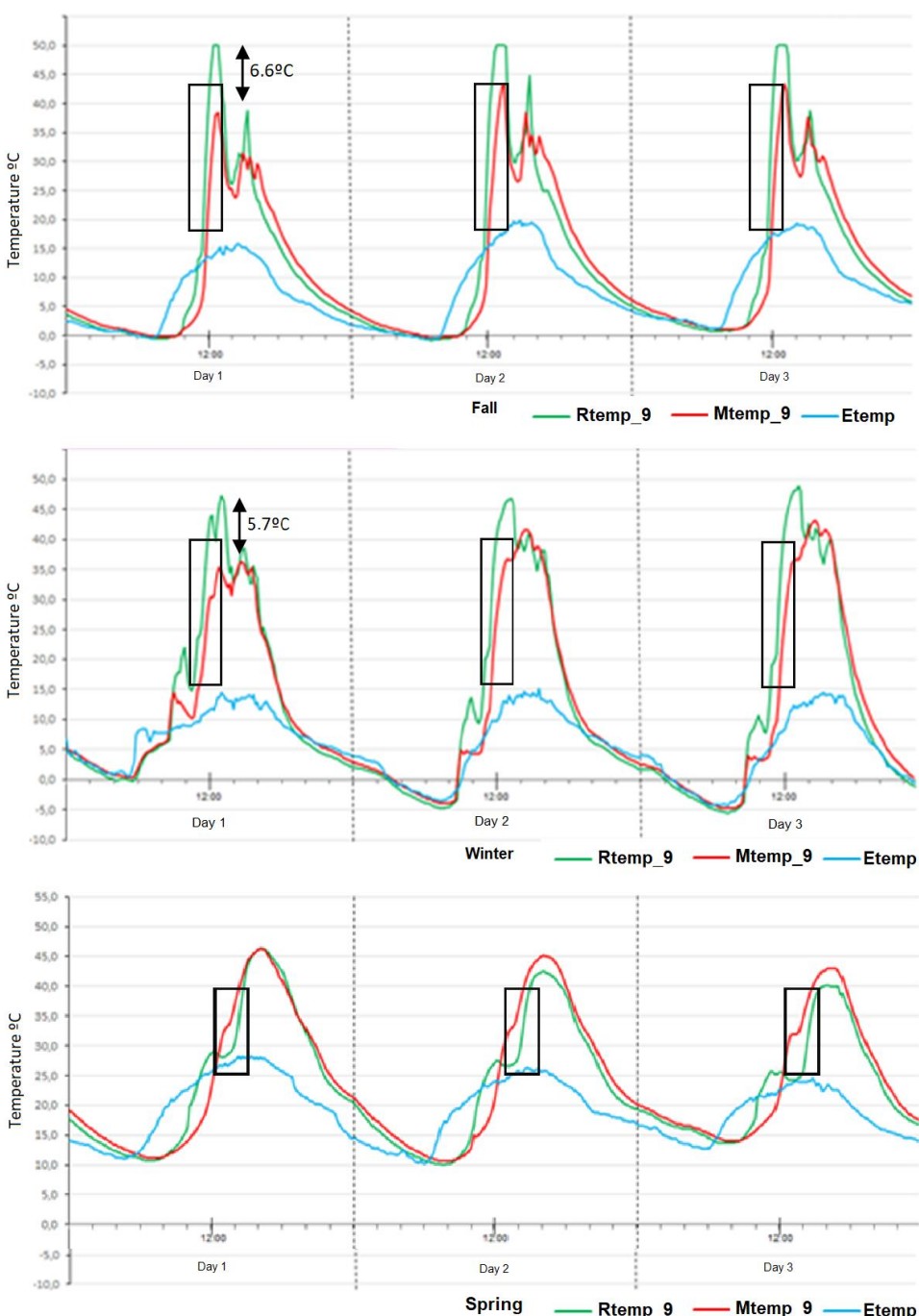

**Figure 8.** Indoor and outdoor ambient temperatures during the autumn, winter, and spring seasons.

## 3.2. Experimental Data of Surface Temperatures

Regarding the exterior surface temperatures in the summer season, the installation of cantilevers facing south favored a decrease in temperature. In the east orientation, both glazings followed the same trend with no major differences in the surface temperatures of both models. In this orientation, the incidence of sunlight on the glazing was lower and the installation of cantilevers was not as beneficial as in the south orientation (Figures 9 and 10).

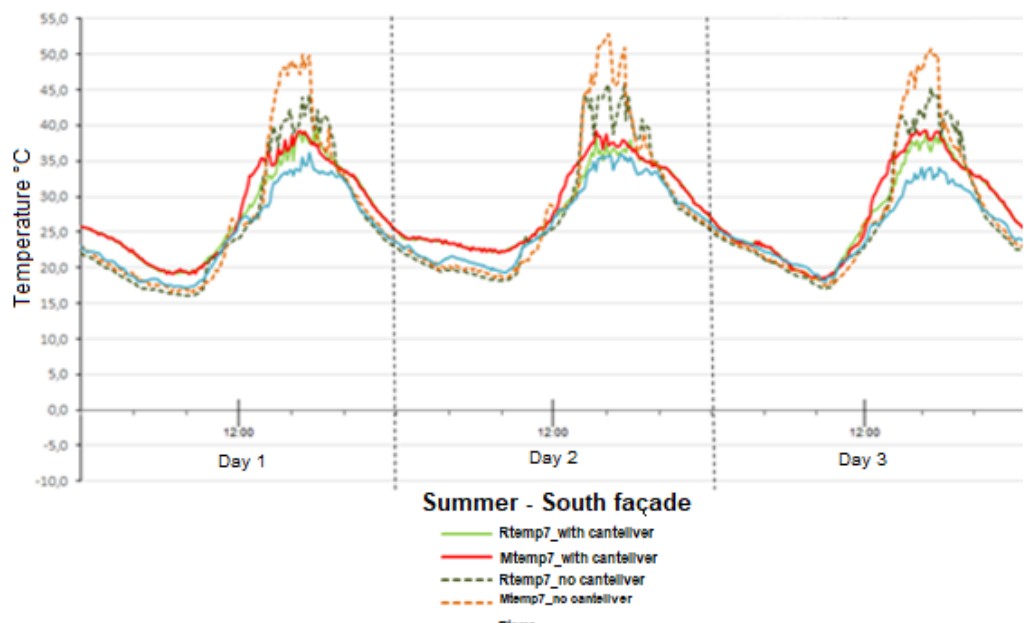

**Figure 9.** Exterior surface temperatures of the south facade during the summer season.

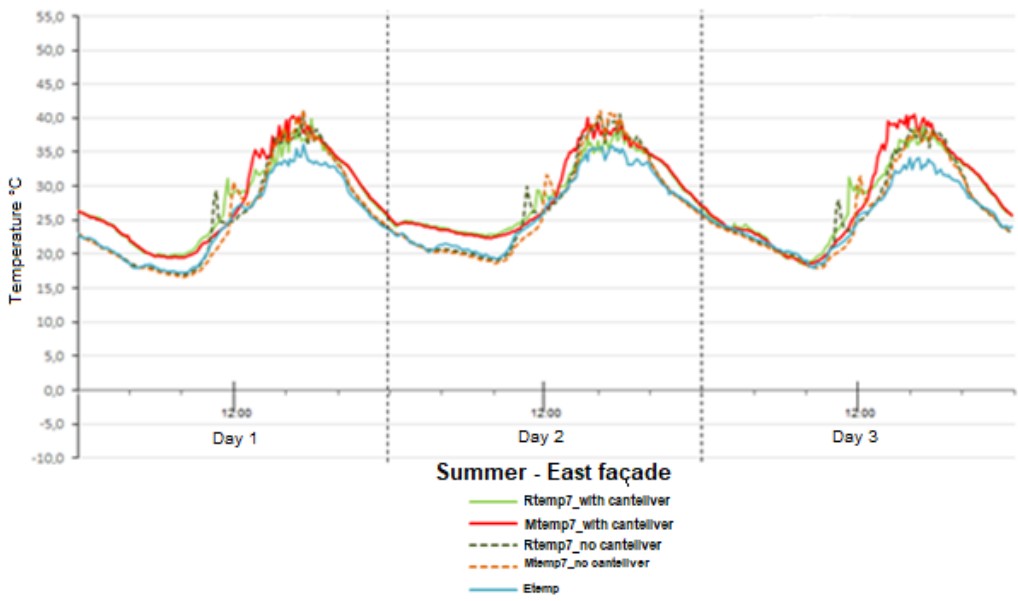

**Figure 10.** Exterior surface temperatures of the east facade during the summer season.

In the winter season in the south orientation, the maximum surface temperature of the double glazing with a solar control sheet was 9.60 °C more than the 6 mm thick single glass.

It should be noted that from the moment the sun began to set, the improved glass model decreased in temperature less than the simple glass model. This was due to the solar control sheet that accumulated all the radiant heat and prevented it from entering the house before gradually releasing it with a drop in temperature.

For the north orientation, double glazing without LCS attained similar temperatures and followed the same trend as the outdoor ambient temperature. The maximum surface temperature of the simple glass model was 4.80 °C more than that of the improved model, representing the reverse situation of the south orientation.

The east and west orientations behaved similarly to the south orientation; however, in the central hours of the day (approximately 12:00–16:00), the temperature increased much

less compared to the south orientation due to the angle of incidence of the solar radiation (Figure 11).

**Figure 11.** Exterior surface temperatures during the autumn, winter, and spring seasons.

The behavior of the average interior surface temperature in the improved model was always higher than the average exterior surface temperature in both orientations and at all weather stations. This was due to the accumulation of heat generated inside the model during the daytime period.

Additionally, it was observed that the south facade that was subjected to direct solar radiation exhibited lower values than the north facade. This indicates that the solar control sheet prevented the passage of heat to the interior (Tables 5 and 6).

**Table 5.** Analysis of average indoor and outdoor surface temperatures for the north orientation.

| Season | Average Outside Surface Temperature (°C) | Average Inside Surface Temperature (°C) | Average Difference between Exterior and Interior Surface Temperatures (°C) |
|---|---|---|---|
| Autumn | 7.59 | 9.34 | −1.75 |
| Spring | 20.53 | 22.97 | −2.44 |
| Winter | 4.97 | 8.01 | −3.04 |
| Summer, cantilever | 27.78 | 28.96 | −1.18 |
| Summer, no cantilever | 25.62 | 27.58 | −1.96 |

**Table 6.** Analysis of average indoor and outdoor surface temperatures for the south orientation.

| Season | Average Outside Surface Temperature (°C) | Average Inside Surface Temperature (°C) | Average Difference between Exterior and Interior Surface Temperatures (°C) |
|---|---|---|---|
| Autumn | 10.37 | 12.00 | −1.63 |
| Spring | 22.88 | 24.83 | −1.95 |
| Winter | 9.75 | 11.70 | −1.95 |
| Summer, cantilever | 28.10 | 29.02 | −0.92 |
| Summer, no cantilever | 27.88 | 29.51 | −1.63 |

### 3.3. Experimental Data on Solar Radiation

The variation in solar radiation is due to the factors that determine the intensity and number of hours of sunshine during the process of the Earth's translation with the sun. Air masses and the associated cloud cover determine the amount of solar radiation that is reflected by clouds, decreasing or preventing it from reaching the Earth's surface.

According to data provided by the State Meteorological Agency (AEMET), the hours of sunshine fluctuate during the different weather seasons within a narrow range of between 11.33–13.66 h of sunshine per day. To be more precise, sunshine is available for 54.17% of the time during the summer, 47.22% during autumn, 51.39% during winter, and 56.94% during spring. The two hours or so difference between the stations causes the peak of maximum irradiance values to be earlier or later depending on which season is being studied.

The maximum radiation measured outside the scale models is reached in the time window between 14:30–17:00. The spring season reached the highest point with 1186 W/m$^2$, followed by summer with a maximum irradiance of 777 W/m$^2$, winter with 312 W/m$^2$, and autumn with an irradiance of 216 W/m$^2$ (Figure 12).

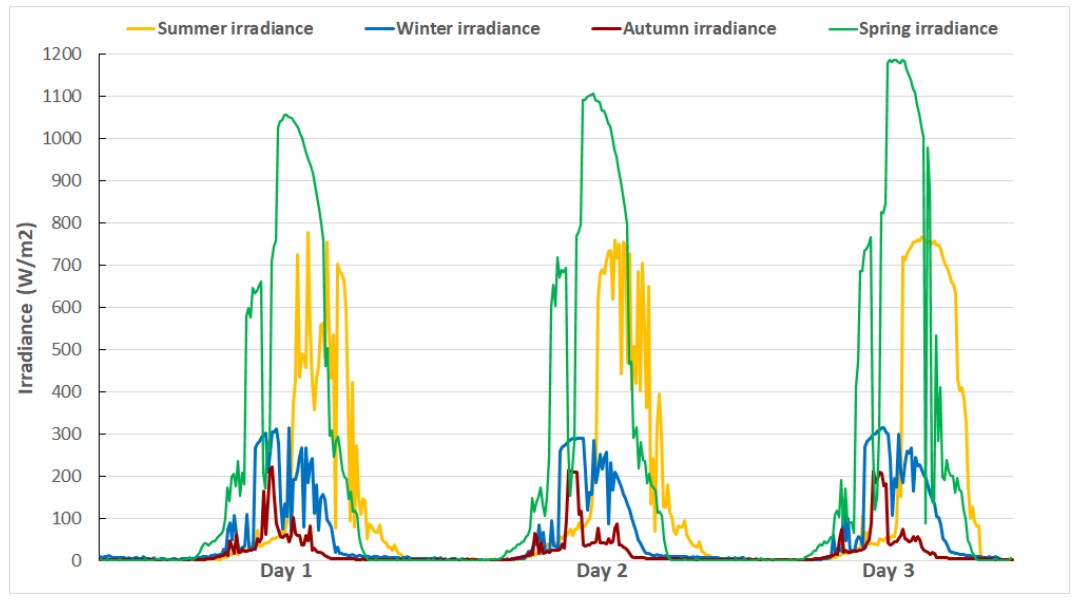

**Figure 12.** Solar radiation during the four weather seasons.

The correlation coefficient between outdoor irradiance and indoor ambient temperature in the scale models is highest in the winter, autumn, and spring months. In these months, the correlation coefficient is above 0.70 but in the summer season it decreases to approximately 0.50 in all cases. The correlation in summer is lower because glazing with

solar control film causes part of the radiation to be reflected to the outside, preventing it from passing into the interior of the home and thus the relationship with the interior temperature is lower.

It is worth noting that the correlation coefficient is always higher (in all seasons) in the case of the improved model. This is due to the fact that the passive protections affect the relationship it may have with the irradiance and the decrease of the indoor temperature (Table 7).

**Table 7.** Analysis of average indoor and outdoor surface temperatures for the south orientation.

| Period | Outdoor Ambient Temperature (°C) | | | Solar Radiation (W/m2) | | | Indoor Ambient Temperature (°C) | | | Correlation Coeff. | Determination Coeff. | |
|---|---|---|---|---|---|---|---|---|---|---|---|---|
| | Max. Temp. | Min. Temp. | Average. Temp. | Max. Irrad. | Min. Irrad. | Average. Irrad. | Max. Temp. | Min. Temp. | Average. Temp. | r (Irrad./ Ind.temp) | $R^2$ (Irrad./ Ind.temp) | |
| Summer without cantilever | 35.1 | 16.9 | 24.8 | 777 | 0 | 144 | 50.0 | 16.8 | 28.6 | 0.48 | 0.23 | Ref. mod. |
| | | | | | | | 50.0 | 17.1 | 28.7 | 0.51 | 0.26 | Imp. mod. |
| Summer with cantilever | | | | | | | 44.4 | 18.5 | 29.1 | 0.45 | 0.20 | Ref. mod. |
| | | | | | | | 43.6 | 18.7 | 29.0 | 0.53 | 0.24 | Imp. mod. |
| Autumn | 19.9 | −0.6 | 7.7 | 216 | 0 | 23 | 50.0 | −0.9 | 12.1 | 0.70 | 0.48 | Ref. mod. |
| | | | | | | | 43.4 | −0.4 | 11.6 | 0.92 | 0.84 | Imp. mod. |
| Winter | 15.1 | −4.3 | 5.6 | 312 | 0 | 66 | 48.9 | −5.5 | 12.2 | 0.76 | 0.58 | Ref. mod. |
| | | | | | | | 43.2 | −4.8 | 11.3 | 0.87 | 0.76 | Imp. mod. |
| Spring | 28.1 | 19.0 | 12.7 | 1186 | 0 | 254 | 46.3 | 10.0 | 23.3 | 0.72 | 0.52 | Ref. mod. |
| | | | | | | | 46.3 | 10.6 | 24.0 | 0.79 | 0.62 | Imp. mod. |

Regarding the outdoor ambient temperature, it is worth noting that it was 48.79% higher in the summer than in the spring. However, average solar radiation was 43.31% higher in the spring than in the summer. Regarding the colder months, the average outdoor ambient temperature in autumn was 27.36% higher than in the winter but the average solar radiation was 65.15% higher in the winter than in autumn.

It follows that there is a poor correlation between irradiance and temperature that demonstrates daytime radiation is not only associated with temperature but also with other transmissivity properties of the atmosphere.

## 4. Conclusions

The results obtained experimentally provide satisfactory scientific support for the studied passive strategies.

The installation of cantilevers in the summer season influenced the interior temperature especially in the central hours of the day, mitigating the increase in temperature and slowing nighttime cooling. The use of a cantilever is favorable for the summer season during which the temperatures in Madrid are high. Conversely, the solar control sheets only partially prevented the passage of heat energy, thus are insufficient for the summer months. In addition, their reflection of energy led to an increase in external surface temperature.

In the remaining seasons, a great difference between the outdoor ambient temperature and the surface temperature was observed especially in the central hours of the day. From the moment the sun began to set, the improved model decreased in temperature less than in the reference model. This was due to the solar control sheet located on the glazing of the improved model that accumulated the radiant heat and prevented its passage to the interior before gradually releasing it as temperatures decreased.

In the winter months during which the temperatures are colder, the glass panels with solar control performed worse. For stations with low outside temperatures, it would be

preferable to use glass panels with low thermal transmittance, thereby lowering the passage of energy between both faces. Accordingly, it can be deduced that glass panels with a solar control film performed better in seasons with warmer temperatures.

It should be noted that the above results are based on the scenarios presented. Other influencing factors such as humidity, wind speed, location, and so on should be taken into account on a case-by-case basis. Therefore, the results should be used with caution. However, these conclusions can be applied to full-scale buildings bearing in mind that this research represents an analysis of single family houses with large glazed surfaces, which is an important factor that new dwellings tend to be prone to.

**Author Contributions:** Conceptualization, P.A.-B. and S.V.-L.; methodology, P.A.-B.; software, C.P.-R.; validation, P.A.-B., S.V.-L. and C.P.-R.; formal analysis, P.A.-B.; investigation, P.A.-B.; resources, S.V.-L.; data curation, C.P.-R.; writing—original draft preparation, S.V.-L.; writing—review and editing, P.A.-B.; visualization, C.P.-R.; supervision, P.A.-B.; project administration, P.A.-B. All authors have read and agreed to the published version of the manuscript.

**Funding:** This research received no external funding.

**Institutional Review Board Statement:** Not applicable.

**Informed Consent Statement:** Not applicable.

**Conflicts of Interest:** The authors declare no conflict of interest.

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
