# Peer review of "Thermal Behavior in Glass Houses through the Analysis of Scale Models"

_sustainability, doi:10.3390/su13147970_

Round 1

Reviewer 1 Report

This manuscript Aguilera-Benito et al. presents monitoring temperature results of two existing scale models of a glass house in Spain. The method used for analysing is reasonable but it needs to be improved. Important standard temperature parameters have been extracted for thermal regulation performance comparisons. I recommend the publication of this manuscript after review some aspects:

- It is considered necessary to analyze the solar radiation together with the temperature data

- Figure 2 shows that there are leftovers of nearby trees on one of the models. Have the leftovers of the environment and those of the models themselves been taken into account in the conclusions?

- It is considered necessary a theoretical section showing how the passive measures taken (cantilever, sheets, etc.) influence the energy behavior

Author Response

Consulte el archivo adjunto.

Reviewer 2 Report

The paper reflects the current trend of larger window openings among households. Authors used an original idea to conduct an experiment using 1:6 scale models to describe, understand and predict behavior of a fully-glazed house. This experiment was extremely time-consuming and the results provide some very interesting insights on this hot topic, however there are multiple shortcomings that lower the overall quality and contribution of the paper. Here is the list of the most important ones I would recommend fixing in order for the paper to be published:

  • Please avoid lumping references in the first two chapters. While this is acceptable in a reasonable extent, nine (line 82) or even ten (line 59) references for one statement are definitely excessive. As for the information on lines 57–59, it is even unclear whether authors mean that these are the “few (that) were based on the thermal analysis of glass envelopes” or some of the 10 references are based on the analysis and some are not.
  • There are multiple important parameters related to the design of the model that are not well explained in the paper:
    • How did you select the scale of the models?
    • Why is the central volume slightly displaced toward the north face?
    • Could you provide more details on the calculations that led to the design of overhangs (solar radiation minimization)?
  • On lines 120 to 122, you mention that “the sensors were positioned according to a previous analysis” and also that your goal was to “address the set objectives”. The reference 38 is used to support this statement is in Spanish and does not seem to be related to the previous analysis. This makes the sentence extremely vague and unclear to a reader.
  • There should be cardinal directions in figures 1 and 4. An attentive reader can deduct the model orientation from the text, but it significantly reduces clarity.
  • The sensor names are not very informative nor consistent through the work (e.g., Rtem_x or Rtemp_x). I would recommend fixing these in all the figures.
  • On lines 126 and 127, you mention “no difference in temperatures between the interior and exterior” for the reference scale model. I would recommend rephrasing this absolute statement to “negligible” or “insignificant” difference.
  • For the comparison of the case with and without a cantilever (chapter 3.1), you mention a deviation of only 2 % - is this based on absolute temperatures or degrees Celsius? Did you measure and consider sun intensity which is probably more important than temperature?
  • In my opinion, the sentence on lines 161 to 164 is not understandable: “In the remaining stations three consecutive days were selected where the average temperature difference was the smallest possible from the average value calculated at each station.” I suspect the word “station” was meant to be “season”.
  • Linear regression might not be the right tool to predict inside temperatures based on outside temperatures. For a given time point, the inside temperature depends on the previous outside temperatures more than it does on the current value. Also, the coefficient of determination should not be confused with probability. It is an indisputable fact that outside and inside temperature are related, therefore the sentence about a “41.4% probability of the exterior temperature influencing the interior temperature” with overhangs is inadequate.
  • The disparity in surface temperature differences between the south and north orientation do not seem to be significant, considering that the influence of humidity and sun exposure on the outside surface temperature reading could be affecting the results. The discussion should be adjusted accordingly.

The main drawback is the lack of discussion on how the results translate to real-life households. For instance, just the fact that the models are not placed on the ground reduces validity of the results. I would be careful using the sentence “we demonstrated the viability of using scale models to analyze energy behavior of air conditioning system”, since there is no information about a model-reality agreement, model limitations or procedure to apply results to different scales including the “life-size” houses. The same applies for air conditioning and, importantly, heating demands as solar radiation minimization might not always be optimal from the energy point of view.

Round 2

Reviewer 1 Report

This work can be a good reference for future related studies, particularly useful because it dealt with real experimental data. . I recommend the publication of this manuscript 

Reviewer 2 Report

I appreciate the authors' responsible approach to the revision. Great job. I can recommend the article for publication.